# Homology-Based Image Processing for Automatic Classification of Histopathological Images of Lung Tissue

**DOI:** 10.3390/cancers13061192

**Published:** 2021-03-10

**Authors:** Mizuho Nishio, Mari Nishio, Naoe Jimbo, Kazuaki Nakane

**Affiliations:** 1Department of Radiology, Kobe University Graduate School of Medicine, 7-5-2 Kusunoki-cho, Chuo-ku, Kobe 650-0017, Japan; 2Division of Pathology, Department of Pathology, Kobe University Graduate School of Medicine, 7-5-1 Kusunoki-cho, Chuo-ku, Kobe 650-0017, Japan; marin@med.kobe-u.ac.jp; 3Department of Diagnostic Pathology, Kobe University Graduate School of Medicine, 7-5-2 Kusunoki-cho, Chuo-ku, Kobe 650-0017, Japan; naoe1123@med.kobe-u.ac.jp; 4Department of Molecular Pathology, Osaka University Graduate School of Medicine and Health Science, Osaka 565-0871, Japan; k-nakane@sahs.med.osaka-u.ac.jp

**Keywords:** pathology image, lung cancer, homology, Betti number, texture analysis, machine learning

## Abstract

**Simple Summary:**

The purpose of this study was to develop a computer-aided diagnosis (CAD) system for automatic classification of histopathological images of lung tissues. Homology-based image processing (HI) was proposed for CAD. For developing and validating CAD with HI, two datasets of histopathological images of lung tissues were used. The private dataset consists of 94 histopathological images that were obtained for the following five categories: normal, emphysema, atypical adenomatous hyperplasia, lepidic pattern of adenocarcinoma, and invasive adenocarcinoma. The public dataset consists of 15,000 histopathological images that were obtained for the following three categories: lung adenocarcinoma, lung squamous cell carcinoma, and benign lung tissue. For the two datasets, our results show that HI was more useful than conventional texture analysis for the CAD system.

**Abstract:**

The purpose of this study was to develop a computer-aided diagnosis (CAD) system for automatic classification of histopathological images of lung tissues. Two datasets (private and public datasets) were obtained and used for developing and validating CAD. The private dataset consists of 94 histopathological images that were obtained for the following five categories: normal, emphysema, atypical adenomatous hyperplasia, lepidic pattern of adenocarcinoma, and invasive adenocarcinoma. The public dataset consists of 15,000 histopathological images that were obtained for the following three categories: lung adenocarcinoma, lung squamous cell carcinoma, and benign lung tissue. These images were automatically classified using machine learning and two types of image feature extraction: conventional texture analysis (TA) and homology-based image processing (HI). Multiscale analysis was used in the image feature extraction, after which automatic classification was performed using the image features and eight machine learning algorithms. The multicategory accuracy of our CAD system was evaluated in the two datasets. In both the public and private datasets, the CAD system with HI was better than that with TA. It was possible to build an accurate CAD system for lung tissues. HI was more useful for the CAD systems than TA.

## 1. Introduction

In 2020, 228,820 new lung cancer cases are projected to occur in the United States [1]; lung cancer is the leading cause of cancer-related deaths in the United States, with almost one-quarter of all cancer deaths being caused by lung cancer. An estimated 606,520 Americans will die from cancer in 2020, with 72,500 male and 63,220 female Americans dying from lung cancer [1].

Currently, histopathological and molecular subtypes are important in lung cancer diagnoses to determine a treatment strategy, and accurate histopathological diagnoses allow clinicians to select targeted treatment options that are specific to each patient. For example, erlotinib (Tarceva; Genentech, South San Francisco, CA, USA) is a tyrosine kinase inhibitor effective in lung cancer patients with mutated epidermal growth factor receptor [2]. Clinicians determine the use of tyrosine kinase inhibitor based on histopathological diagnoses of the mutated epidermal growth factor receptor. Generally, immunohistochemistry is used for the diagnosis of the mutated epidermal growth factor receptor.

Digital pathology systems [3,4] have improved over the years and are now capable of producing high-resolution histopathological images. Using digital pathology systems, histopathological assessments can be performed using a computer display rather than a light microscope. In addition, this system has enabled computer-aided diagnosis (CAD) for histopathological diagnosis. Currently, CAD is being used for detection and diagnosis in several medical fields [5,6,7], and CAD has the potential to improve the speed and accuracy of histopathological diagnoses of lung cancer [3].

CAD frequently utilizes machine learning to improve its diagnostic accuracy. In order to use medical images in CAD, image feature extraction is required for machine learning. For evaluation of tumor aggressiveness, tumor heterogeneity is an important factor [8,9]. In CAD of cancers, texture analysis is frequently used for image feature extraction to assess tumor heterogeneity [8,9].

In recent years, homology-based image processing has been increasingly used [10,11,12,13,14,15,16,17]. For example, Nishio et al. showed that homology-based image processing was useful for estimating the risk of lung cancer [15], and Nakane et al. showed that colon cancer could be accurately segmented on histopathological images using homology-based methods [14]. In homology-based methods, Betti numbers are import metrics for image feature extraction. These numbers are calculated from binarized images obtained from medical images (please refer to Figure 2 of [13] and Appendix A of [17] for the calculation of Betti numbers). In the current study, it was assumed that Betti numbers obtained with homology-based image processing were useful for evaluation of tumor heterogeneity in image feature extraction. 

The purpose of this study was to develop a CAD system for the automatic classification of histopathological images. To the best of our knowledge, image feature extraction of histopathological images of lung tissue has not been performed using homology-based image processing, and the performance of CAD has not been appraised when homology-based image processing has been used. For the purpose of this study, private and public datasets were used. In the proposed method, the histopathological images were automatically classified using image features extracted based on the homology method and several machine learning algorithms. For comparison with the proposed method, conventional texture analysis was used for image feature extraction.

## 2. Materials and Methods

This retrospective study was approved by the institutional review board of our institution (permission number: B200033); the requirement for acquiring informed consent was waived.

### 2.1. Private Dataset

In the private dataset, ninety-four histopathological images of lung tissue were obtained from lung surgery specimens. They belonged to five categories of lung tissue (normal, emphysema, atypical adenomatous hyperplasia (AAH), lepidic pattern of adenocarcinoma (LP), and invasive adenocarcinoma (AC)), consisting of 20 normal, 20 emphysema, 23 AAH, 19 LP, and 12 AC images. The histopathological diagnosis of the 94 images was confirmed by two board-certified pathologists (M.N. and N.J.). These histopathological images were obtained by means of hematoxylin and eosin staining. The image resolution of the 94 images was 1600 × 1200 pixels with RGB channels at 100× magnification (the magnification of the objective lens being 10×). Figure 1A–E show representative histopathological images of the five categories, respectively.

For developing and evaluating the CAD system, the 94 histopathological images of the private dataset were randomly divided into a training set with 50 images, a validation set with 20 images, and a testing set with 24 images. Because the number of images in the private dataset was small, image patches were extracted from the images for each of the three sets. Ten image patches with image resolution 1024 × 1024 pixels were randomly extracted from one histopathological image. In addition, vertical and horizontal flipping were randomly applied to the image patches as in data augmentation of deep learning [6]. Finally, a training set with 500 image patches, a validation set with 200 image patches, and a testing set with 240 image patches were used for the CAD system.

### 2.2. Public Dataset

The public dataset (LC25000) contains 25,000 color images with five classes of 5000 images each [18]. All images are 768 × 768 pixels in size. From LC25000, 15,000 histopathological images of three classes of lung tissue (lung adenocarcinoma, lung squamous cell carcinoma, and benign lung tissue) were selected. Figure 2A–C show representative histopathological images of the three categories, respectively.

As in the private dataset, the 15,000 histopathological images of LC25000 were divided into a training set with 9000 images, a validation set with 3000 images, and a test set with 3000 images. The image patch extraction was not used for the public dataset.

### 2.3. Outline of CAD System

Figure 3 shows an outline of the CAD system for the private dataset. Except for the output, the same processing was performed for the public dataset. The RGB images were fed into the CAD system and then the image features were extracted. A machine learning algorithm classified the image based on the extracted features. To train the machine learning algorithm of the CAD system and optimize parameters of the CAD system, image features of the training and validation sets were used, respectively. Finally, image features of the testing set were used for assessing the performance of the CAD system. The programming language used for the development of the CAD system was Python (version 3.7, http://www.python.org/ (accessed on 13 November 2020)).

### 2.4. Image Feature Extraction

To perform homology-based image processing, Betti numbers (b0 and b1) were calculated for the histopathological images with RGB channels. Figure 4 shows an outline of the Betti number calculation process for histopathological images. To calculate the Betti numbers, a grayscale image converted from the RGB images was prepared. Because Betti numbers are calculated using a binarized image in which each pixel can have two values (0 and 1), the grayscale image was binarized before calculating the Betti numbers. For binarization, thresholding was performed using predefined pixel values. The binarized images obtained via thresholding were processed using our in-house homology software to calculate the Betti numbers. The process of calculating the Betti numbers from binarized images has been described elsewhere [13,14,15,16,17]. Briefly, in a two-dimensional binarized image, b0 (the zero-dimensional Betti number) is the number of connected components in the image, and b1 (the one-dimensional Betti number) is the number of one-dimensional or “circular” holes in the image. For the predefined pixel value of thresholding, 0, 5, 10, …, 245, 250, and 255 were used. For multiscale analysis, the image resolution was changed in calculating the Betti numbers. In the private dataset, the image resolutions of 1024 × 1024, 512 × 512, 256 × 256, and 128 × 128 pixels were used for multiscale homology-based image processing. In the public dataset, the image resolutions of 768 × 768, 384 × 384, 192 × 192, and 96 × 96 pixels were used. Image features of the Betti numbers at different image resolutions were concatenated, based on our multiscale homology-based image processing. A schematic illustration of the multiscale homology-based image processing is shown in Figure 5.

For the conventional method, image feature extraction was performed using texture analysis via PyRadiomics (version 3.0, https://pyradiomics.readthedocs.io/en/latest/ (accessed on 14 November 2020)) [19]. Texture analysis was performed on a grayscale image converted from the original RGB image. The target of texture analysis was the entire image. The image feature names of the texture analysis are listed in the Appendix A (Appendix A). Briefly, 18, 23, 16, 16, 14, and 5 image features were calculated for First Order, Gray Level Co-occurrence Matrix, Gray Level Run Length Matrix, Gray Level Size Zone Matrix, Gray Level Dependence Matrix, and Neighboring Gray Tone Difference Matrix, respectively. The multiscale analysis was also performed for texture analysis.

### 2.5. Preprocessing of Image Features and Machine Learning

In the current study, scikit-learn (version 0.23.2, https://scikit-learn.org/stable/ (accessed on 14 November 2020)) was used for both preprocessing of the image features and the machine learning algorithms [20]. After the image feature extraction, preprocessing of the image features was performed. For the preprocessing, the standardization of image features and feature selection were utilized. The mean and standard deviation of each feature was used for the standardization of image features using the sklearn.preprocessing.StandardScaler class. After the standardization, the feature selection was performed using the sklearn.feature_selection.SelectKBest class and sklearn.feature_selection.f_classif function, where the number of selected features was set to 20% of the original image features. Both the feature standardization and the feature selection were optional. The image features with or without the preprocessing were fed into the machine learning algorithms. The machine learning algorithms included (0) perceptron, (1) logistic regression, (2) kNN, (3) support vector machine with linear kernel, (4) support vector machine with radial basis function kernel, (5) decision tree, (6) random forest, and (7) gradient tree boosting. For gradient tree boosting, xgboost (version 1.2.0, https://xgboost.readthedocs.io/en/latest/ (accessed on 14 November 2020)) was used [21]. These machine learning algorithms were trained with the image features of training set and their default hyperparameters provided by the implementation of scikit-learn and xgboost. 

### 2.6. Performance Evaluation

Performance evaluation was performed using the multicategory classification accuracy obtained in the testing set. For determining the optimal CAD, the validation accuracy was calculated for all possible combinations of normalization, feature selection, image resolution, and machine learning algorithms. For both homology-based image processing and texture analysis, single-scale and multiscale analyses were performed. All combinations of image resolutions (1024 × 1024, 512 × 512, 256 × 256, and 128 × 128 pixels for the private dataset, and 768 × 768, 384 × 384, 192 × 192, and 96 × 96 pixels for the public dataset) were used for multiscale analysis. 

## 3. Results

Table 1, Table 2, Table 3 and Table 4 and Appendix A show prediction results of the CAD systems for the private and public datasets. In each entry of Appendix A, the most accurate result and its corresponding algorithm were selected among the eight machine learning algorithms. In the optimal machine learning algorithm of Table 1, Table 2, Table 3 and Table 4 and Appendix A, 0–7 represents perceptron, logistic regression, kNN, support vector machine with linear kernel, support vector machine with radial basis function kernel, decision tree, random forest, and gradient tree boosting, respectively. 0 and 1 in the normalization and feature selection of these tables represent “without preprocessing” and “with preprocessing”, respectively.

Appendix A show validation accuracies of the CAD systems with homology-based image processing and texture analysis for all possible combinations in the private dataset, respectively. Table 1 and Table 2 show the validation and testing accuracies of the optimal CAD systems with homology-based image processing and texture analysis selected from Appendix A, respectively. According to Table 1 and Table 2, the testing accuracy of the optimal CAD with the homology-based image processing (78.33%) was better than that with the texture analysis (70.83%). Random forest and logistic regression were used in Table 1 and Table 2, respectively. Because single-scale analysis was used in the entries of Table 1 and Table 2, the usefulness of multiscale analysis was limited for the private dataset.

Appendix A show validation accuracies of the CAD systems with homology-based image processing and texture analysis for all possible combinations in the public dataset, respectively. Table 3 and Table 4 show validation and testing accuracies of the optimal CAD systems with homology-based image processing and texture analysis selected from Appendix A, respectively. According to Table 3 and Table 4, the best testing accuracy of the optimal CAD with the homology-based image processing (99.43%) was better than that with the texture analysis (99.33%). Gradient tree boosting was frequently used in Table 3 and Table 4. Because no entry of single-scale analysis was found in Table 3 and Table 4, multiscale analysis was useful in the public dataset.

Figure 6 and Figure 7 show the confusion matrices between the ground truth and prediction, which were obtained with the optimal CAD systems for the private and public datasets, respectively.

## 4. Discussion

The results of this study indicate that it is possible to construct an accurate CAD system by using homology-based image processing for the multicategory classification of lung tissue (normal, emphysema, AAH, LP, and AC in the private dataset, and lung adenocarcinoma, lung squamous cell carcinoma, and benign lung tissue in the public dataset). Our results show that the accuracy of the multicategory classification with homology-based image processing was better than that with texture analysis in the private and public datasets.

Classification of AAH, LP, and AC is important because it affects patient prognosis and survival [22]. For instance, the identification of pure LP has been shown to have excellent prognoses for patients with stage I lung cancer [23]. However, accurate classification of such patterns can be challenging [24]. Because the classification accompanies the subjective nature of pathologists, interobserver variability of the pathologists’ diagnosis can be problematic. Our CAD system might be helpful in solving this problem.

To our knowledge, few studies have used machine learning or deep learning to predict the histological subtype classification of lung tissue [3]. One study performed a six-category classification of histologic patterns in lung adenocarcinoma and benign tissue (lepidic, acinar, papillary, micropapillary, solid, and benign) [25]. Another study performed the five-category classification (solid, micropapillary, acinar, cribriform, and nontumor) in lung adenocarcinoma and nontumor tissue [26]. Compared with these two studies, our novelty is that our CAD system distinguished AAH from the other four categories. In addition, while these two studies used deep learning, our study used machine learning.

To evaluate the efficacy of homology-based image processing in the large dataset, the public dataset obtained from LC25000 was used in this study. The results for the public dataset show that homology-based image processing was more useful than conventional texture analysis in the classification between lung adenocarcinoma, lung squamous cell carcinoma, and benign lung tissue. 

In this study, it was assumed that homology-based image processing was useful for evaluating tumor heterogeneity in the CAD system of lung cancer. Because our results show that CAD with homology-based image processing was more accurate than that with texture analysis, our assumption was validated. One major advantage of homology-based image processing over texture analysis is topological invariance [14]. Because of this property, Betti numbers are not changed by continuous transformation. It is speculated that in the CAD system with homology-based image processing, topological invariance makes image features more robust, compared with texture analysis.

The multiscale analysis improved the accuracy of both homology-based image processing and texture analysis for the public dataset. It is speculated that because the image resolution is essential information for image classification, multiscale analysis was useful for the two methods of image feature extraction. On the other hand, the usefulness of multiscale analysis was not clear for the private dataset. This might be caused by an imbalance between dataset size and number of image features in the multiscale analysis. Further study is needed to establish the usefulness of the multiscale analysis in homology-based image processing. 

According to Figure 6, classification between AAH and LP was difficult in our optimal CAD system. One major reason for this result is the size of the private dataset. Generally, machine learning and deep learning yield relatively poor performance for small datasets. Although we used patch-level accuracy for mitigating the effect of the small dataset, we could not avoid deterioration in the classification between AAH and LP. To overcome this problem, a larger dataset should be used.

Our study has several limitations. First, the private dataset was small. For mitigating the effect of the small dataset, patch-level accuracy was evaluated in the private dataset. In addition, a public dataset was also used in this study. Second, no external validation was performed. Overfitting of our CAD system may have occurred in the external validation. For future studies, we need to investigate the effectiveness of our CAD systems using datasets obtained from other affiliations. Third, the subtype classification of adenocarcinoma was not fully investigated. Of adenocarcinoma, minimally invasive adenocarcinoma and adenocarcinoma in situ [22] were not considered for the private dataset. Classification between AAH, minimally invasive adenocarcinoma, adenocarcinoma in situ, and invasive adenocarcinoma should be performed in future development of our CAD system. Fourth, classification of adenocarcinoma between lepidic predominant, acinar predominant, papillary predominant, micropapillary predominant, etc. was not performed. This classification should be investigated in future study. Fifth, we did not compare our CAD system with that with deep learning. Because the private dataset was small, it was speculated that the performance of the CAD system with deep learning might be low in the private dataset. Therefore, we did not use deep learning in this study. Sixth, although several studies investigated CAD systems for the prognosis, survival, and genetic features of lung cancers [27,28,29], we did not predict them in the current study. Because the classification between LP and AC is directly related to prognosis and survival of lung cancer [23], we believe that our CAD system is useful for evaluating the prognosis and survival of lung cancer.

## 5. Conclusions

It was possible to build an accurate CAD system for the automatic classification of lung tissue. Homology-based image processing was more useful for CAD systems than conventional texture analysis.

## Figures and Tables

**Figure 1 cancers-13-01192-f001:**
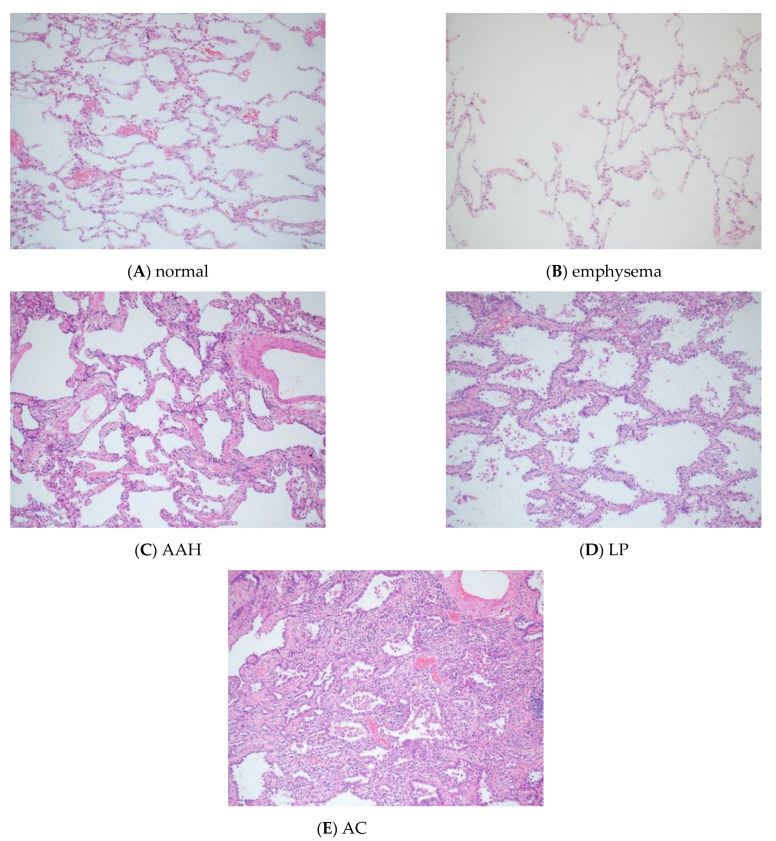
Representative histopathological images of (**A**) normal, (**B**) emphysema, (**C**) AAH, (**D**) LP and (**E**) AC. The magnification is 100× (the magnification of the objective lens being 10×). AAH, atypical adenomatous hyperplasia; LP, lepidic pattern of adenocarcinoma; AC, invasive adenocarcinoma.

**Figure 2 cancers-13-01192-f002:**
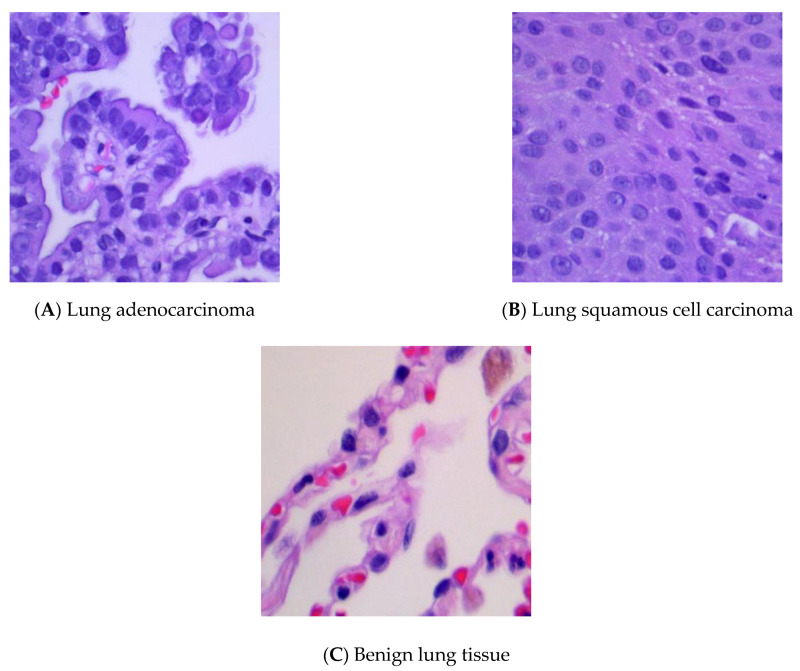
Representative histopathological images of (**A**) lung adenocarcinoma, (**B**) lung squamous cell carcinoma and (**C**) benign lung tissue.

**Figure 3 cancers-13-01192-f003:**
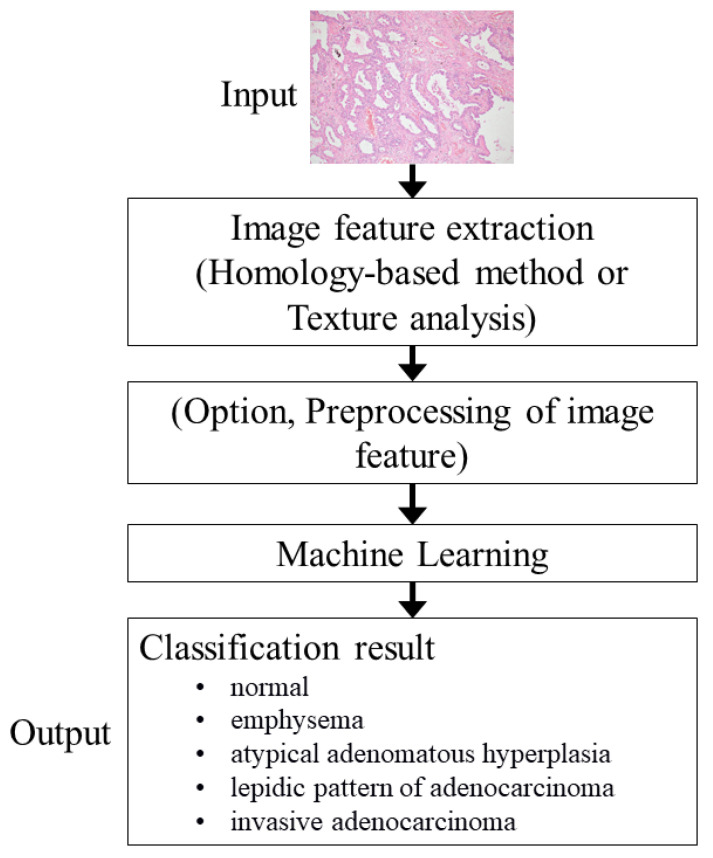
Outline of the CAD system in private dataset. Note: Except the output, the same processing was performed for the public dataset. CAD, computer-aided diagnosis.

**Figure 4 cancers-13-01192-f004:**
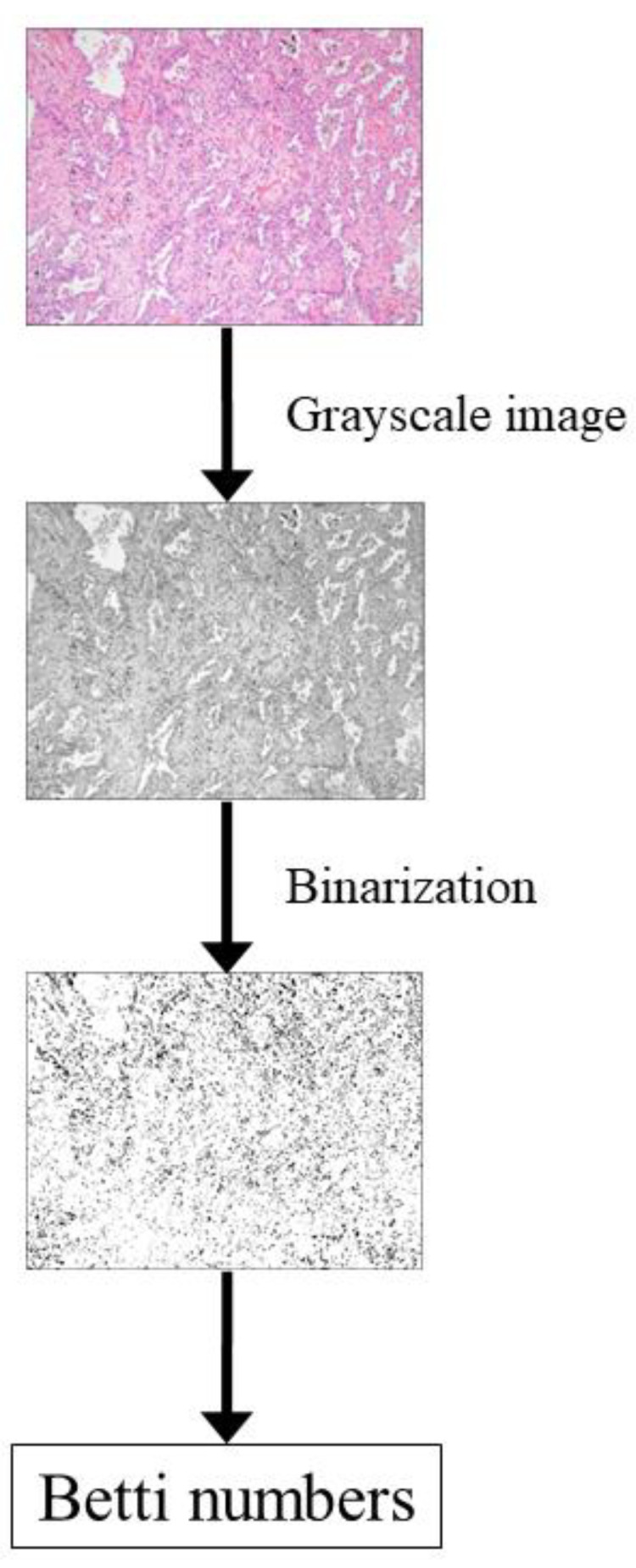
Outline of the process of calculating Betti numbers.

**Figure 5 cancers-13-01192-f005:**
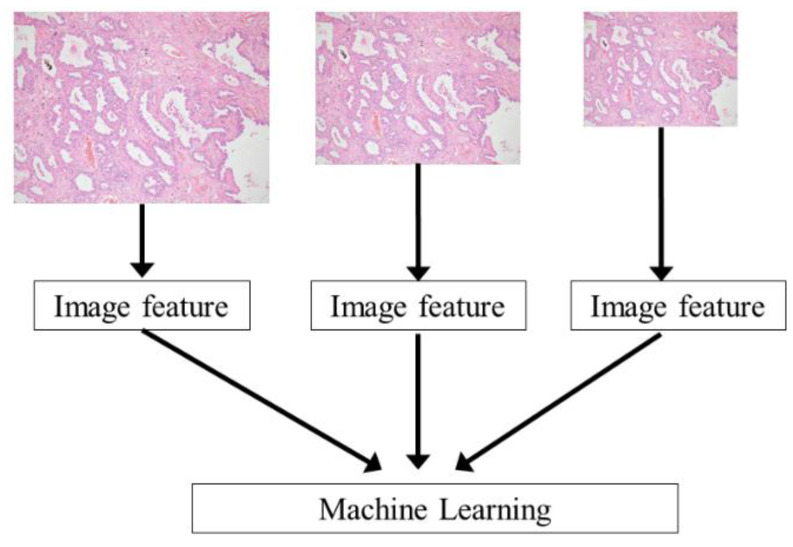
Schematic illustration of the multiscale homology-based image processing.

**Figure 6 cancers-13-01192-f006:**
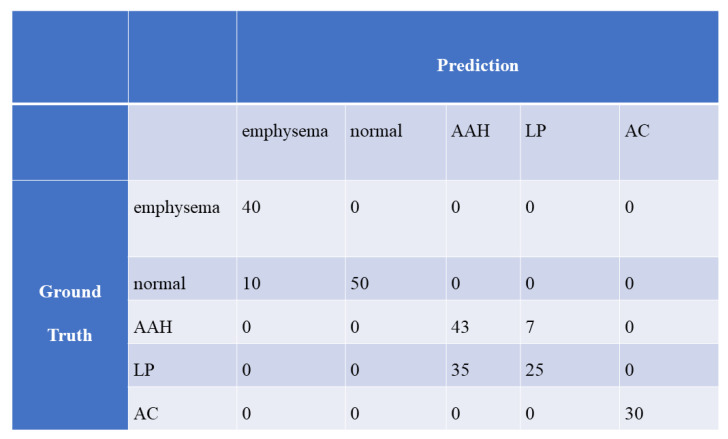
Confusion matrix between the ground truth and prediction obtained with the optimal CAD system for the private dataset.

**Figure 7 cancers-13-01192-f007:**
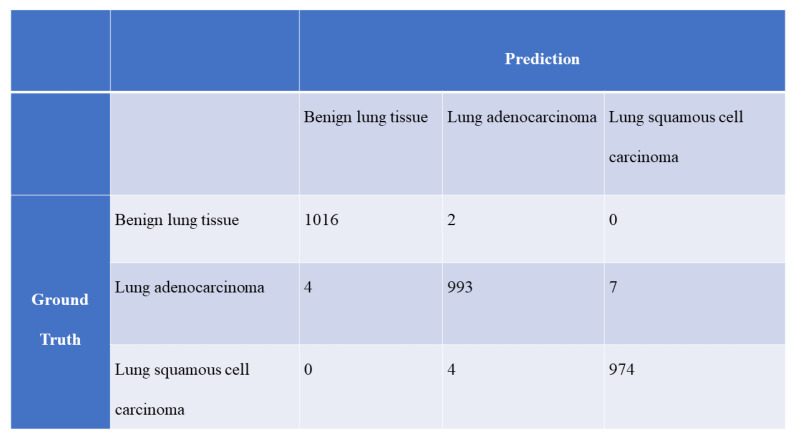
Confusion matrix between the ground truth and prediction obtained with the optimal CAD system in the public dataset.

**Table 1 cancers-13-01192-t001:** Validation and testing five-category accuracies of the optimal CAD with homology-based image processing in the private dataset. Note: The optimal CAD was selected based on the validation accuracies of Appendix A.

Normalization	Feature Selection	Image Resolutions (Pixels)	Validation Accuracy	Testing Accuracy	Optimal Machine Learning Algorithm
0	1	256 × 256	0.9000	0.7833	6

**Table 2 cancers-13-01192-t002:** Validation and testing five-category accuracies of the optimal CAD with texture analysis in the private dataset. Note: The optimal CAD was selected based on the validation accuracies of Appendix A.

Normalization	Feature Selection	Image Resolutions (Pixels)	Validation Accuracy	Testing Accuracy	Optimal Machine Learning Algorithm
0	1	1024 × 1024	0.8650	0.7083	1

**Table 3 cancers-13-01192-t003:** Validation and testing three-category accuracies of the optimal CAD with texture analysis in the public dataset. Note: The optimal CAD was selected based on the validation accuracies of Appendix A.

Normalization	Feature Selection	Image Resolutions (Pixels)				Validation Accuracy	Testing Accuracy	Optimal Machine Learning Algorithm
1	0	1024 × 1024	512 × 512	256 × 256		0.9927	0.9940	2
0	0	512 × 512	256 × 256	128 × 128		0.9927	0.9923	7
0	0	1024 × 1024	512 × 512			0.9927	0.9920	7
0	0	1024 × 1024	512 × 512	256 × 256	128 × 128	0.9927	0.9943	7
1	0	512 × 512	256 × 256	128 × 128		0.9927	0.9923	7
1	0	1024 × 1024	512 × 512			0.9927	0.9920	7
1	0	1024 × 1024	512 × 512	256 × 256	128 × 128	0.9927	0.9943	7

**Table 4 cancers-13-01192-t004:** Validation and testing three-category accuracies of the optimal CAD with texture analysis in the public dataset. Note: The optimal CAD was selected based on the validation accuracies of Appendix A.

Normalization	Feature Selection	Image Resolutions (Pixels)				Validation Accuracy	Testing Accuracy	Optimal Machine Learning Algorithm
0	0	1024 × 1024	512 × 512	256× 256	128 × 128	0.9923	0.9933	7
1	0	1024 × 1024	512 × 512	256× 256	128 × 128	0.9923	0.9933	7

## Data Availability

The public dataset is available in a publicly accessible repository. Please see [18]. The private dataset is not available because of regulation of privacy.

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
