# Peer review of "Homology-Based Image Processing for Automatic Classification of Histopathological Images of Lung Tissue"

_cancers, 2021, doi:10.3390/cancers13061192_

Round 1

Reviewer 1 Report

1. It has been widely accepted that deep learning models could extract complicated image features compared with traditional feature extraction methods. The features extracted from neural network can represents both low level features (color, shape, edge, texture, etc.) and high level features (spatial structure, super-pixel, etc.). A pretrained/fine-tuned backbone can automatically including the two main categories of features highlighted in this paper (TA and HI). With these advanced features, classification tasks with deep neural network backbone could achieve much higher classification accuracy. What's the advantage of choosing traditional HI and TA over modern deep learning based method? (when the paper focusing on accuracy instead of interpretation)

2. The categorization of the 5 subtypes in the private dataset is not convincing. From Figure 1, the 5 subtypes could be easily identified on low level magnification by simply counting the ratio of background vs. tissue and the percentage of holes. (No doubt that Betti numbers could achieve very high accuracy on this task). From the view of pathologists, identifying complicated subtypes is more meaningful: lepidic predominant, acing predominant, papillary predominant, micro-papillary predominant, etc. Please considering these subtypes to evaluate the performance of the presented CAD system.

3. CV accuracy is not a proper evaluation metric. It's strongly recommended to split your data into: training dataset, validation dataset and another independent testing dataset. Run CV or whatever on the training dataset for best parameters and hyper-parameters. Then provide the performance on validation dataset to decide the best model. Further more, apply the pretrained best model on the independent testing dataset to see the generalization power. For the private dataset, as the sample size is small, please provide patch level accuracy instead of whole slide accuracy. For the LC25000, split into train/val/test is a must as the dataset is large enough.

4. As the presented system choose to use more interpretable features with HI and TA, what's the significance of each features? (p-value, coefficients, etc). Are there any features positive/negative correlated to some subtypes?

5. Image features are correlated with prognosis, survival, genetic features. So providing further analysis with genetic data and longitudinal data would be strongly recommended. As the methodology has no novelty, presenting strong biological incites are important for a high impact journal.

Author Response

Reviewer #1

The authors would like to thank the Reviewer 1 for his/her useful comment to improve the overall quality of our manuscript. Our point-by-point response to the reviewer’s comment is shown below.

#1-1

It has been widely accepted that deep learning models could extract complicated image features compared with traditional feature extraction methods. The features extracted from neural network can represents both low level features (color, shape, edge, texture, etc.) and high level features (spatial structure, super-pixel, etc.). A pretrained/fine-tuned backbone can automatically including the two main categories of features highlighted in this paper (TA and HI). With these advanced features, classification tasks with deep neural network backbone could achieve much higher classification accuracy. What's the advantage of choosing traditional HI and TA over modern deep learning based method? (when the paper focusing on accuracy instead of interpretation)

In general, deep neural networks are computationally expensive, while texture analysis (TA) and homology-based image processing (HI) are less computationally expensive. In addition, deep neural networks have the problem of low accuracy on small datasets such as our private dataset. For these reasons, we focused on TA/HI and machine learning in this study.

The creator of LC25000 (public dataset) used LC25000 and deep neural networks for classification between squamous cell carcinoma and adenocarcinoma. His result shows that his deep neural network achieved 94% accuracy in the testing set. While classification targets are different between his trial and our study, our CAD with HI is better than the deep neural network of LC25000 creator. Please refer to the following URL: https://medium.com/analytics-vidhya/sub-classifying-lung-cancer-with-tensorflow-2-and-keras-616353e59e5e

#1-2

The categorization of the 5 subtypes in the private dataset is not convincing. From Figure 1, the 5 subtypes could be easily identified on low level magnification by simply counting the ratio of background vs. tissue and the percentage of holes. (No doubt that Betti numbers could achieve very high accuracy on this task). From the view of pathologists, identifying complicated subtypes is more meaningful: lepidic predominant, acing predominant, papillary predominant, micro-papillary predominant, etc. Please considering these subtypes to evaluate the performance of the presented CAD system.

Our private dataset consists of five categories of lung tissue (normal, emphysema, atypical adenomatous hyperplasia, lepidic pattern of adenocarcinoma, and invasive adenocarcinoma). Therefore, it is impossible for us to investigate CAD for classifying adenocarcinoma between lepidic predominant, acinar predominant, papillary predominant, micropapillary predominant, and etc. In the revised manuscript, this point is clarified as limitation.

Fourth, classification of adenocarcinoma between lepidic predominant, acinar pre-dominant, papillary predominant, micropapillary predominant, and etc. was not per-formed. This classification should be investigated as future study.

#1-3

CV accuracy is not a proper evaluation metric. It's strongly recommended to split your data into: training dataset, validation dataset and another independent testing dataset. Run CV or whatever on the training dataset for best parameters and hyper-parameters. Then provide the performance on validation dataset to decide the best model. Further more, apply the pretrained best model on the independent testing dataset to see the generalization power. For the private dataset, as the sample size is small, please provide patch level accuracy instead of whole slide accuracy. For the LC25000, split into train/val/test is a must as the dataset is large enough.

Following this recommendation, we divided each of the two datasets into training/validation/testing sets and performed the experiment again. The training set was used for training machine learning algorithms, and the validation set was used to select the optimal CAD. We used the optimal CAD to evaluate the multi-category accuracy in the testing set. The results are described in the Results section of the main text and in the Supplementary material. Following the comment of Reviewer 2, the results of the validation set are shown in the Supplementary material. The main text contains only the results for the optimal CAD. Because the results in the revised manuscript show limited effectiveness of the multi-scale analysis, several parts of the paper have been revised to address this issue.

#1-4

As the presented system choose to use more interpretable features with HI and TA, what's the significance of each features? (p-value, coefficients, etc). Are there any features positive/negative correlated to some subtypes?

As shown in the manuscript, the feature selection was performed using the sklearn.feature_selection.SelectKBest class. In this study, SelectKBest class performed the feature selection based on the results of sklearn.feature_selection.f_classif function. The f_classif function calculated ANOVA F-value. Therefore, statistical significance between single feature and class label was considered in feature selection of this study.

For sklearn.feature_selection.SelectKBest class and sklearn.feature_selection.f_classif function, please refer to the following URLs.

https://scikit-learn.org/stable/modules/generated/sklearn.feature_selection.SelectKBest.html

https://scikit-learn.org/stable/modules/generated/sklearn.feature_selection.f_classif.html#sklearn.feature_selection.f_classif

#1-5

Image features are correlated with prognosis, survival, genetic features. So providing further analysis with genetic data and longitudinal data would be strongly recommended. As the methodology has no novelty, presenting strong biological incites are important for a high impact journal.

Because our datasets do not include data of prognosis, survival, and genetic features, we could not investigate CAD for prognosis, survival, and genetic features. In the revised manuscript, this point is clarified as limitation.

Sixth, although several studies investigated CAD systems for prognosis, survival, and genetic features of lung cancers [27–29], we did not predict them in the current study. Because the classification between LP and AC is directly related with prognosis and survival of lung cancer [23], we believe that our CAD system is useful for evaluating the prognosis and survival of lung cancer.

However, we believe that use of homology-based image processing for developing CAD of lung tissue is novel as the methodology.

Reviewer 2 Report

This is an interesting study that I think can be improved in several ways. The clinical problem is of interest. The main conclusion is that homology based image processing outperforms texture analysis. The authors are realistic in their self criticism in the final paragraph.

Introduction: The introduction could be improved by mentioning the importance in genetics/mutational analysis in determining treatment. I do not agree that rapid diagnosis is that important and this is over played. The terms homology based and texture based deserve more explanation, particularly as the paper is aimed at practising clinicians and pathologists. If it is not aimed at this audience then some of the criticisms methodologically mentioned later become more important. Other than that both techniques exist it is not made sufficiently clear why this study is important for machine learning rather than for diagnosis.

Methods: These are explained with clear diagrams. A significant concern which to be fair the authors raise themselves is using a primary/private dataset of only 94 images representing five different conditions. There is considerable heterogeneity, even within normal and certainly in adenocarcinoma, with only 12 cases contributing to one condition. It is difficult to envisage how this can be adequate, and the authors also acknowledge there is no external validation set. It would be valuable to address this problem.

Results: The tables add little and perhaps could be summarised and placed with supplementary. Figure 2a from the public dataset is probably micropapillary adenocarcinoma again highlighting the importance of tumour heterogeneity which the authors also acknowledge. In figure 6 the predictive value is not great for LP: 24% were incorrectly predicted. Of itself this is not surprising but highlights the importance of using a larger dataset.

Discussion: It would be interesting and important for the authors to discuss more why the results show that homology based image processing is superior to textural analysis, in particular to comment on issues like heterogeneity, likely differences between laboratories in real world setting, and how different laboratory processing may influence parameters. Only in lines 264 and 265 is this mentioned. There is a paragraph at the end explaining candidly limitations of the study and how some were mitigated. 

Author Response

Reviewer #2

This is an interesting study that I think can be improved in several ways. The clinical problem is of interest. The main conclusion is that homology based image processing outperforms texture analysis. The authors are realistic in their self criticism in the final paragraph.

The authors would like to thank the Reviewer 2 for his/her useful comment to improve the overall quality of our manuscript. Our point-by-point response to the reviewer’s comment is shown below.

#2-1

Introduction: The introduction could be improved by mentioning the importance in genetics/mutational analysis in determining treatment. I do not agree that rapid diagnosis is that important and this is over played. The terms homology based and texture based deserve more explanation, particularly as the paper is aimed at practising clinicians and pathologists. If it is not aimed at this audience then some of the criticisms methodologically mentioned later become more important. Other than that both techniques exist it is not made sufficiently clear why this study is important for machine learning rather than for diagnosis.

Thank you for the insightful comment. Based on this comment, Introduction section was modified as follows. 

Currently, histopathological and molecular subtypes are important in lung cancer diagnoses to determine a treatment strategy, and accurate histopathological diagnoses allows clinicians to select targeted treatment options that are specific to each patient. For example, Erlotinib (Tarceva; Genentech, South San Francisco, Calif) is a tyrosine kinase inhibitor effective in lung cancer patients with mutated epidermal growth factor receptor [2]. Clinicians determine use of tyrosine kinase inhibitor based on histopathological diagnoses of the mutated epidermal growth factor receptor.

CAD frequently utilizes machine learning to improve its diagnostic accuracy. In order to use medical images in CAD, image feature extraction is required for machine learning. For evaluation of tumor aggressiveness, tumor heterogeneity is an important factor [8,9]. In CAD of cancers, texture analysis is frequently used for image feature extraction to assess tumor heterogeneity [8,9].

In recent years, homology-based image processing has been increasingly used [10–17]. For example, Nishio et al. showed that homology-based image processing was useful for estimating the risk of lung cancer [15], and Nakane et al. showed that colon cancer could be accurately segmented on histopathological images using homology-based methods [14]. In homology-based methods, Betti numbers are import metrics for image feature extraction. These numbers are calculated from binarized image obtained from medical image (please refer to Figure 2 of [13] and S1 Figure of [17] for the calculation of Betti numbers). In the current study, it was assumed that Betti numbers obtained with homology-based image processing was useful for evaluation of tumor heterogeneity in image feature extraction. 

#2-2

Methods: These are explained with clear diagrams. A significant concern which to be fair the authors raise themselves is using a primary/private dataset of only 94 images representing five different conditions. There is considerable heterogeneity, even within normal and certainly in adenocarcinoma, with only 12 cases contributing to one condition. It is difficult to envisage how this can be adequate, and the authors also acknowledge there is no external validation set. It would be valuable to address this problem.

Thank you for the comment. Following the Reviewers 1 and 2’s comment, the development and evaluation of our CAD was modified in the revision. In the private dataset, the 94 images were divided into training set with 50 images, validation set with 20 images, and testing set with 24 images. In addition, patch-level accuracy was used for the development and evaluation of our CAD. Please refer to the following paragraph in the revised manuscript.

For developing and evaluating the CAD system, the 94 histopathological images of the private dataset were randomly divided into training set with 50 images, validation set with 20 images, and testing set with 24 images. Because the number of images in the private dataset was small, image patches were extracted from the images for each of the three sets. Ten image patches with image resolution 1024×1024 pixels were randomly extracted from one histopathological image. In addition, vertical and horizontal flipping were randomly applied to the image patches as in data augmentation of Deep learning [6]. Finally, training set with 500 image patches, validation set with 200 image patches, and testing set with 240 image patches were used for the CAD system.

#2-3

Results: The tables add little and perhaps could be summarised and placed with supplementary. Figure 2a from the public dataset is probably micropapillary adenocarcinoma again highlighting the importance of tumour heterogeneity which the authors also acknowledge. In figure 6 the predictive value is not great for LP: 24% were incorrectly predicted. Of itself this is not surprising but highlights the importance of using a larger dataset.

Because Tables 1, 2, 5, and 6 of original submission are huge, these Tables has been moved to Supplementary material (Table S2-5) in the revised manuscript. In the revised manuscript Tables 1-4 contains only the results for the optimal CAD.

In this study, it was assumed that Betti numbers obtained with homology-based image processing was useful for evaluation of tumor heterogeneity in image feature extraction. This point is highlighted in Introduction section. Please refer to Introduction section of the revised manuscript.

As the Reviewer 1 appointed, it is difficult to differentiate between AAH and LP in our optimal CAD system. We clarified the difficulty in classification between AAH and LP in the revised manuscript.

According to Figure 6, classification between AAH and LP was difficult in our optimal CAD system. One major reason for this result is the size of the private dataset. Generally, machine learning and Deep learning yield relatively poor performance in small dataset. Although we used patch-level accuracy for mitigating effect of the small dataset, we could not avoid deterioration in the classification between AAH and LP. To overcome this problem, larger dataset should be used.

#2-4

Discussion: It would be interesting and important for the authors to discuss more why the results show that homology based image processing is superior to textural analysis, in particular to comment on issues like heterogeneity, likely differences between laboratories in real world setting, and how different laboratory processing may influence parameters. Only in lines 264 and 265 is this mentioned. There is a paragraph at the end explaining candidly limitations of the study and how some were mitigated.

Thank you for the comment. Based on the recommendation, we modified Discussion section as follows.

In this study, it was assumed that homology-based image processing was useful for evaluating tumor heterogeneity in the CAD system of lung cancer. Because our results show that CAD with homology-based image processing was more accurate than that with texture analysis, our assumption was validated. One major advantage of homology-based image processing over texture analysis is topological invariant [14]. Because of the property, Betti numbers are not changed by continuous transformation. It is speculated that in the CAD system with homology-based image processing, topological invariant makes image features more robust, compared with texture analysis.

Round 2

Reviewer 2 Report

The comments have been addressed thoroughly. Mention is made in the introduction that histopathology diagnosis gf mutation is made. This is pushing a little far: it is histological examination of immunohistochemistry not of an H&E image. This is minor and can easily be changed.

Author Response

Reviewer #2

The authors would like to thank the Reviewer 2 for his/her useful comment to improve the overall quality of our manuscript. Our point-by-point response to the reviewer’s comment is shown below.

#2-1

The comments have been addressed thoroughly. Mention is made in the introduction that histopathology diagnosis gf mutation is made. This is pushing a little far: it is histological examination of immunohistochemistry not of an H&E image. This is minor and can easily be changed.

Thank you for the comment. Following the Reviewers 2’s comment, Introduction section was modified as follows. 

Clinicians determine use of tyrosine kinase inhibitor based on histopathological diagnoses of the mutated epidermal growth factor receptor. Generally, immunohistochemistry is used for the diagnosis of the mutated epidermal growth factor receptor.
